# Substantial Antigenic Drift in the Hemagglutinin Protein of Swine Influenza A Viruses

**DOI:** 10.3390/v12020248

**Published:** 2020-02-23

**Authors:** Pia Ryt-Hansen, Anders Gorm Pedersen, Inge Larsen, Charlotte Sonne Kristensen, Jesper Schak Krog, Silke Wacheck, Lars Erik Larsen

**Affiliations:** 1National Veterinary Institute, Technical University of Denmark, Kemitorvet Building 204, DK-2800 Kongens Lyngby, Denmark; 2Dpt. of Veterinary and Animal Sciences, University of Copenhagen, Grønnegårdsvej 2, DK-1870 Frederiksberg C, Denmark; inge@sund.ku.dk (I.L.); lael@sund.ku.dk (L.E.L.); 3Department of Health Technology, Section for Bioinformatics, Technical University of Denmark, Kemitorvet Building 208, DK-2800 Kongens Lyngby, Denmark; agpe@dtu.dk; 4SEGES, Danish Pig Research Centre, Agro Food Park 15, DK-8200 Aarhus, Denmark; csk@seges.dk; 5Statens Serum Institut, Artillerivej 5, DK-2300 Copenhagen S, Denmark; JSKR@ssi.dk; 6Ceva Santé Animale 10 Avenue de la Ballastière, 33500 Libourne, France; silke.wacheck@ceva.com

**Keywords:** swine influenza A virus, genetic drift, antigenic drift, molecular clock, immune escape variants, selection, enzootic infections, substitution rate, hemagglutinin, sows

## Abstract

The degree of antigenic drift in swine influenza A viruses (swIAV) has historically been regarded as minimal compared to that of human influenza A virus strains. However, as surveillance activities on swIAV have increased, more isolates have been characterized, revealing a high level of genetic and antigenic differences even within the same swIAV lineage. The objective of this study was to investigate the level of genetic drift in one enzootically infected swine herd over one year. Nasal swabs were collected monthly from sows (*n* = 4) and piglets (*n* = 40) in the farrowing unit, and from weaners (*n* = 20) in the nursery. Virus from 1–4 animals were sequenced per month. Analyses of the sequences revealed that the hemagglutinin (HA) gene was the main target for genetic drift with a substitution rate of 7.6 × 10^−3^ substitutions/site/year and evidence of positive selection. The majority of the mutations occurred in the globular head of the HA protein and in antigenic sites. The phylogenetic tree of the HA sequences displayed a pectinate typology, where only a single lineage persists and forms the ancestor for subsequent lineages. This was most likely caused by repeated selection of a single immune-escape variant, which subsequently became the founder of the next wave of infections.

## 1. Introduction

Novel influenza A viruses (IAV) can develop through two different mechanisms; genome reassortment (antigenic shift) and gradual accumulation of mutations (antigenic drift). Genome reassortment occurs as a consequence of the segmented genome of IAV, when RNA segments originating from different subtypes/strains are mixed during assembly of progeny virions, leading to the formation of new subtypes/strains that may have novel antigenic properties [1,2]. Antigenic drift is a much slower process where the error prone RNA polymerase causes misincorporation of nucleotides during genome replication [3,4]. Mutations in coding regions of the viral genome are either synonymous or non-synonymous. Non-synonymous mutations occurring in immunogenic epitopes can undergo positive selection driven by host immunity, and may lead to the virus escaping e.g. neutralizing antibodies. The major epitopes of IAV, also termed antigenic sites, are located on the globular head of the hemagglutinin (HA) molecule, which is encoded by the HA1 domain. Several antigenic sites have been identified for both the H1 and H3 subtypes [5,6,7,8,9,10]. For humans, positive selection in these sites has been documented [11,12,13,14], and as little as one mutation in an antigenic site has been shown to affect the vaccine effectiveness [15,16]. Human influenza vaccines are therefore, evaluated twice a year to prevent mismatches between vaccine strain and circulating strains [17]. 

The rate of antigenic drift of swine IAV (swIAV) has generally been believed to be much lower than that of human IAV, mainly due to the short lifespan of pigs and the acute nature of the infection historically seen in pig herds, which limits the impact of pre-existing immunity [3,18,19,20,21,22]. Consequently, the swine influenza vaccines are updated less frequently [23]. Previous studies on the antigenic drift of swine hemagglutinin of the H1 or H3 subtypes, has mainly focussed on the global or national evolution [19,24,25,26,27,28,29]. For the H1 subtype in swine the nucleotide substitution rates have been estimated to range between 1.9–4.4 × 10^−3^ per site per year [11,30,31,32,33], whereas the nucleotide substitution rate of the swine H3 subtype has been documented to be as high as 6 × 10^−3^ per site per year [34]. While the highest rates are comparable to that of human H1 subtypes, the selection pressure expressed as the ratio of nonsynonymous to synonymous mutations has been found to be lower in swine compared to humans [11,22,35,36]. However, a very recent study revealed similar rates of synonymous and non-synonymous nucleotide substitutions within the swine H1 lineages 1B and 1C compared to that of human IAV, when investigating the genetic drift of swIAV sampled between 2003–2015 in Germany [37]. 

Over the past ten years, the understanding of swIAV circulation in swine herds has changed and it is now recognized that an infection with swIAV is likely to result in an enzootic infected herd [38,39,40,41,42,43,44,45]. This is probably a consequence of increasing herd sizes, which provides a continuous flow of naïve piglets [46,47]. The possible impact that herd level persistence of IAV might have on antigenic drift over time in a specific population of pigs, has, to the best of our knowledge, never been investigated. However, we believe it is highly important to get an increased understanding of antigenic drift occurring within single herds, as it can help explain the high genetic diversity within swIAV lineages documented in large-scale investigations and surveillance programs [25,26,27,29,48,49]. Importantly, if positive selection comparable to that observed in human IAV occurs in swine IAV, the possible effects on herd immunity and vaccination, should be taken into consideration when designing vaccines and evaluating swIAV control programs. We here report the results of a repeated cross-sectional study where we investigated the dynamics and viral evolution within a single sow herd over a one-year period. 

## 2. Materials and Methods

### 2.1. Herd Description

The herd consisted of 480 sows and stables for 2000 nursery pigs. The farrowing stables were divided into two units and had no sectioning between different age groups, with weekly farrowings. The farrowing unit was cleaned once a year, without the use of disinfectants. At four weeks of age, piglets were weaned into a heated nursery. The nursery stables contained seven separate rooms with separate airflow, and all rooms were cleaned and disinfected between batches. The nursery pigs were housed in the nursery, until they were sold at approx. 30 kilos. Gilts were recruited internally (annual replacement rate of approx. 50%) and were subjected to eight weeks of quarantine from 12 weeks-of-age. Thus, no pigs were introduced from an outside source into the herd during the study period. According to the Danish Specific Pathogen Free program [50], the herd was free from infection with *Mycoplasma hyopneumoniae*, *Actinobacillus pleuropneumoniae* serotype, 2, 6, and 12, PRRSv type 1 and 2, *Brachyspira hyodysenteriae*, *Pasteurella multocida*, *Sarcoptes scabiei* var. *Suis,* and *Haematopinus suis*. The herd used a low number of nursing sows and minimized cross fostering of piglets. No vaccination against IAV was performed. The herd experienced recurrent respiratory symptoms in both the farrowing and nursery unit, and tested positive for IAV in July 2017, where the subtype “H1avN2sw” was diagnosed by full genome sequencing. 

### 2.2. Study Design

Nasal swabs were collected monthly from November 2017 to October 2018. The nasal swabs were obtained from 20 piglets from four one-week-old litters (five piglets per litter), 20 piglets from four three-week-old litters (five piglets per litter) and 20 pigs from four pens with five-week-old nursery pigs (five pigs per pen). In addition, nasal swabs were collected from each sow of the one-week-old litters, and the parity of the sows was recorded. In total 64 nasal swabs were collected each month corresponding to 768 nasal swabs obtained over the full year. The individual samples and sequences were given an ID ranging from F1–F12 according to which month they were sampled, F1 being the first month (November 2017) and F2 being the second month etc. Moreover, the sample ID also included the age of the pigs, W1, W3, and W5 indicating week 1, 3, and 5, respectively. In addition, sequences obtained previously in July 2017, was included in the genetic analysis and named “W00_W1_01”.

### 2.3. Sampling

Nasal swabs were collected from the sows and piglets using a large or small rayon dry swab (Medical Wire, UK), respectively. The swabs were inserted into each nostril and turned 360 degrees. Thereafter, the swab was inserted in a tube containing 1ml Sigma Virocult media (Medical Wire, Corsham, UK) and kept at 2–8 °C for maximum 24 hours before being processed. 

### 2.4. Coughing Index

For each litter/pen, included in the sampling, a coughing index was calculated, as described in a previous study [45]. Briefly, the coughing index was calculated by dividing the total number of coughs and sneezes with the number of pigs in the litter/pen multiplied by the time observed (three minutes). 

### 2.5. Pooling and RNA Extraction

All the nasal swabs from pigs were initially pooled prior to extraction. Subsequently, the two most positive pools of each sampling time was identified, and the individual nasal swabs of the pools was subjected to RNA extraction and real-time RT PCR, to identify samples for viral isolation and sequencing. All nasal swabs were mixed using a Vortexer, and 100 µL was extracted for the pool. All individual samples (excluding the samples from sows) were pooled litter-wise, with five nasal swabs in each pool. The pools were mixed and centrifuged. Subsequently, 200 µL suspension was transferred to a new tube containing 400 µL RLT-buffer (QIAGEN, Copenhagen, Denmark) with 2-Mercaptoethanol (Merck, Darmstadt, Germany). Subsequently, the RNA was extracted using the RNeasy mini kit (QIAGEN, Copenhagen, Denmark) automated on the QIAcube (QIAGEN, Copenhagen, Denmark) according to large sample protocol version 2.

### 2.6. Real-Time RT PCR

In order to determine if a pool was positive for IAV, a previously published real-time RT PCR assay targeting the matrix gene of IAV [51] was adopted. Briefly, the published primers and the OneStep RT-PCR kit (QIAGEN, Copenhagen, Denmark) was used for the PCR mix, which was subsequently run on the Rotor-Gene Q (QIAGEN, Copenhagen, Denmark) using the following program: 50 °C, 30 min; 95 °C, 15 min; cycling 45× (95 °C for 10 s, 60 °C for 20 s, 64 °C for 1 sec, 68 °C for 1 sec, 72 °C for 30 s). All samples were tested in duplicates and the pool was only considered positive if both samples gave a positive result and had a Ct value <36. All positive samples with a Ct value <31 were tested to determine the IAV subtype using the previously published multiplex real-time RT PCR assay [52] with the modifications described in a previous study [45] and run on the Rotor-Gene Q (QIAGEN, Copenhagen, Denmark). The individual samples of two most positive pools of each sampling were also tested. Positive individual samples were selected for viral isolation and sequencing.

### 2.7. Viral Isolation and NGS

The monthly nasal swab with the lowest Ct value was selected for viral isolation. The nasal swabs were first subjected to sterile filtration using a 0.45 µM Millex-HP Millipore filter (Merck, Darmstadt, Germany) and then grown in Madin-Darby Canine Kidney (MDCK) cells under the conditions as described in a previous study [53]. After incubation, the RNA was extracted from the supernatant of each cell isolate using the same method as described above, however performed manually. Subsequently, the RNA was subjected to PCR amplification of each IAV segment and prepared for sequencing on the Illumina MiSeq platform using the methods described in a previous study [53]. 

### 2.8. HA and NA Amplification and Sanger Sequencing

Additional individual nasal swabs, which had Ct values <31, of each sampling were subjected to HA and NA PCR amplification and subsequent Sanger sequencing using the same methods as described in a previous study [53]. 

### 2.9. Consensus Sequence Generation

The determination of consensus nucleotide and amino acid sequences based on the Illumina and Sanger sequencing data was done as previously described [53] using the program CLC genomics Workbench version 11.0.1 (Aarhus, Denmark) and CLC main workbench version 8 (Aarhus, Denmark), respectively. 

### 2.10. Characterization of the Herd swIAV Strain

The nucleotide and amino acid consensus sequences of each of the eight gene segments were aligned using the MUSCLE algorithm [54] in CLC main workbench version 8. The sequences of the alignments were compared using the pairwise comparison tool. In addition, the lineage of each gene segment was determined by aligning the respective sequences with contemporary swIAV sequences obtained in the Danish swine IAV surveillance program and subsequently neighbor-joining trees were constructed. Furthermore, HA amino acid sequences were annotated for known antigenic sites (Sa, Sb, Ca1, Ca2, and Cb) [5,6,8,55,56], B-cell epitopes [57,58] and T-cell epitopes [59,60,61,62], which were then manually checked for variation. Potential asparagine-linked glycosylation sites of the HA protein was predicted by the program NetNGlyc 1.0 Server [63]. 

### 2.11. Molecular Clock Analysis and Positively Selected Sites 

Neighbor joining trees were constructed for each of the eight gene segments using the CLC main workbench version 8 software (Aarhus, Denmark). The eight gene segments obtained from the diagnostic sample of the same herd sampled approx. 4 months earlier (July 2017), were included in all analyses and used as an outgroup. The resulting tree, including information on sampling dates, was subsequently checked for the presence of a temporal signal (i.e., whether nucleotide changes accumulate proportionally to elapsed time) using the program TempEST [64]. Thereafter, the software package BEAST2 version 2.5.2 [65] was used to determine the substitution rate of each of the eight gene segments as previously described [66]. Briefly, the substitution model was specified to be HKY with gamma distributed rates over sites, with a strict clock model, and using tip dates (sampling dates). The chain length was set to 10,000,000 with a log every 1000, and the MCMC was run twice. The program BEAUti [65] was used to set up the analysis with all priors. Summaries of results and checking of MCMC convergence was done using the program Tracer, version 1.7.1 [67]. 

The program CODEML of the PAML package [68] was used to identify positively selected sites in all 8 genome segments as described in a previous study [66]. Briefly, we did this by comparing the fits of CODEML’s substitution models 1a (M1a) and 2a (M2a) (NSsites = 1 and 2). These substitution models include parameters for the ratio between the rate of non-synonymous substitutions per non-synonymous sites and the rate of synonymous substitutions per synonymous site (the dN/dS ratio, also indicated by ω). A dN/dS ratio above one indicates positive selection (there are more amino acid changing substitutions then expected). M1a includes two categories of codons—some under negative selection (dN/dS ratio < 1) and some codons where mutations are neutral (dN/dS ratio = 1). The model M2a includes three categories of codons—the same two as M1a plus an additional category of codons under positive selection (dN/dS ratio > 1). If M2a fits a dataset significantly better than M1a, then there is evidence of positive selection in some codons (and the identity of these codons is also found during model fitting). The fit of each model was compared using the Akaike Information Criterion (AIC) and likelihood ratio tests [69,70]. In addition, the average dN/dS ratio (global ω ratio) of all HA sequences was also estimated using CODEML (NSsites = 0). We also used CODEML to map codon (and the derived amino acid) mutations to branches in the HA phylogeny. Specifically, we used the M2a model (NSsites = 2) with RateAncestor = 1 to infer the most probable mapping of mutations to branches (mutational mappings are output in the result file “rst” along with ancestral reconstructions). The inferred amino acid changes were subsequently manually plotted on the HA clock-based phylogenetic tree (described below).

The program MrBayes [71] was used for reconstructing clock-based phylogenetic trees using codon-based substitution models, allowing simultaneous estimation of clock rates and detection of positively selected sites for the HA and NA gene segments. This method was described in a previous study [66]. The data analysis was performed using two parallel runs for 3.000.000 generations with a sample frequency of 600. The phylogenetic tree was inferred in a Bayesian framework and with MCMC sampling of posterior probabilities. We used the program Tracer version 1.7.1 to check convergence of MCMC runs [67]. Phylogenies were visualized using FigTree version 1.4.4, and the R-packages “tidyverse”, “treeio”, and “ggtree” in RStudio (version 0.97.551) [72,73,74,75,76,77].

### 2.12. Statistics

Student’s t-test was used to investigate if the average coughing index was significantly different between the IAV positive and IAV negative litters/pens. A chi-squared test was used to evaluate if 1st parity sows were more likely to be IAV positive compared to older sows, and the same test was also used to test if IAV positive sows were more likely to have an IAV positive litter compared to the IAV negative sows. All calculations were done using the GraphPad Software [78]. A likelihood ratio chi-squared test was used to test if M2a fit the data significantly better than M1a (indicating the presence of positively selected codons) [79]. Statistical significance was considered when the *p*-value was below 0.05. 

## 3. Results

### 3.1. Presence of Enzootic IAV

The results of real-time RT PCR targeting the matrix gene revealed that IAV was present at all monthly samplings. Some variations were however observed between months: For example, very few litters/pens tested positive at F11 (September) and the Ct values of the samples did not allow for sequencing (Table 1 and Figure 1). The results of the test of the pooled samples of each month, showed that 60% of the one-week old litters, 69% of the three-week old litters and 60% of the pens with five-week old weaners tested positive over the entire study period. In total, 16 of 48 (33%) sows tested positive for IAV in the nasal swabs over the study period. The majority of IAV positive sows also had a positive litter (14/16). The prevalence of IAV positive litters from IAV positive sows (88%), was significantly higher than the prevalence of IAV positive litters from IAV negative sows (50%) (*p* = 0.03). For first parity sows, seven of 15 (47%) tested positive for IAV in nasal swabs as opposed to only nine of 33 (27%) of the ≥2nd parity sows. However, this difference was not significant (*p* value = 0.32). 

The percentage of swIAV positive samples at week 1, 3, and 5 were based on pooled samples, whereas samples from sows were tested individually.

### 3.2. Correlation between IAV and the Coughing Index

For each age group (weeks 1, 3, and 5) the average coughing index (CI) of the IAV positive and negative litters/pens, was calculated (Table 2). Overall, the mean CI was significantly higher in litters/pens that included at least one pig testing positive for IAV in nasal swabs (*p* value = 0.03). No significant differences were discovered within the individual age groups, but a tendency towards a higher coughing index in the IAV positive litters was most evident at week 1 (*p* value = 0.07). 

### 3.3. Herd Strain Characterization

In total, ten full genome sequences were obtained from cultured isolates based on individual monthly samples. However, it was not possible to obtain viral isolates from F7 and F11. Full genome sequences from one sample obtained in July 2017 were additionally included. Moreover, 19 HA and NA sequences were obtained from RNA of individual nasal swabs of all samplings except from F11. Thus, in total 30 HA and 29 NA sequences were generated, including 1-4 sequences from each sampling time (except F11).

The herd swIAV isolated throughout the study was of the H1N2 subtype, with a HA gene of Eurasian avian-like origin, and a NA gene of the swine adapted Hong Kong H3N2 origin. All gene segments of the internal gene cassette were of Eurasian avian-like origin. The HA gene segments had a pairwise nucleotide sequence identity ranging between 98.6–100% and, similarly, the NA gene segment had a pairwise nucleotide sequence identity ranging between 98.5–100%. The M, NP, PA, PB1, and PB2 gene segments had a pairwise nucleotide sequence identity ranging between 99.2–100% respectively, whereas the NS gene segments had a pairwise nucleotide sequence identity ranging between 97.5–100%. All sequences are available in NCBI Genbank with the following accession numbers: MN990039-MN990079 (Sanger sequences) and MN990117-MN990192 (NGS including internal genes).

### 3.4. Phylogenetic Analysis and Substitution Rates

The TempEst analysis revealed that all eight gene-segments, but especially the HA gene (correlation coefficient 0.95), showed association between genetic divergence through time and sampling dates indicating that a phylogenetic molecular clock-analysis (using BEAST and MrBayes) was suitable for the sequences (Table 3). Using BEAST the nucleotide substitution rate for the HA segment was estimated to be 7.6 × 10^−3^ substitutions/site/year, corresponding to 13 nucleotide substitutions per year for the entire gene (which is 1698 nucleotides long in this dataset). Estimated substitution rates were also high in the NA and NS segments (6.9 × 10^−3^ and 5.7 × 10^−3^ substitutions/site/year respectively), while the remaining segments had substitution rates ranging from 1.1 to 2.9 × 10^−3^ substitutions/site/year.

Interestingly, the phylogenetic tree based on the HA sequences displays the same imbalanced (so-called “comb-like” or pectinate) topology as that which is typical for human influenza trees spanning multiple years. The main feature of this type of topology is the repeated bottlenecks where only a single lineage persists and forms the ancestor for subsequent lineages (Figure 2). This is most likely caused by repeated selection of a single immune-escape variant that becomes the founder of the next wave of infection. The phylogenetic tree based on NA sequences had a somewhat comb-like topology also (Figure 3), however, with fewer bottleneck events (although firm conclusions in this regard is made difficult by greater uncertainty about branching pattern, and the resulting high level of polytomies in the tree).

### 3.5. Positive Selection

The program CODEML from the PAML package was used to test if positive selection was present in the eight gene segments. The results showed that the M2a model (indicating the presence of positive selection) fitted the HA sequences significantly better, whereas this was not the case for the remaining genes (Table 3). Estimates of dN/dS ratios for individual codons in the HA gene, under the M2a model, strongly indicated the presence of positive selection at position 536 (numbering from first methionine) in the HA2 part of the gene, which encodes the stalk region. Further analysis of the HA amino acid alignment showed that a mutation from tryptophan to arginine at position 536 was present in four pigs at F4, F6 and F8 (F4_W1_13, F6_W1_P16, F8_W1_16, F8_W1_17). Interestingly, this specific position is located in a B-cell epitope identified in the A(H1N1)pdm09 virus and in a T-cell epitope identified among human seasonal H1N1 (Table 4). The average dN/dS ratio of the HA sequences was estimated to be 0.19, while the value for codon 536 was estimated at 1.6, supporting that this position was under positive selection. Analysis of dN/dS ratios for individual codons using MrBayes identified several additional sites having a dN/dS ratio below 1, but significantly higher (p < 0.05) than the average dN/dS ratio (0.19) (Table 4). The majority of these mutations were present in the HA1 part of the HA gene and included seven mutations in known antigenic sites and several of the positions were in other known B- or T-cell epitopes. Further investigation of HA sequences identified seven mutations (D125N, P137L, K155R, V216D, E222K, H283Y, and I404F), that all showed a clear temporal pattern, where the given mutation became established at one time point and remained in all the following sequences until the end of the study.

## 4. Discussion

This study documented IAV persistence within a swine herd over a one-year period, supporting the increasing number of studies that have shown that IAV may persists within the herd [38,39,40,41,42,43,44,45]. Furthermore, the study confirms previous studies documenting a lack of seasonality of swIAV infections [28,80]. IAV was abundantly present in the one-week old litters, which in addition was the age group that showed the highest average viral load. These findings support the results of a pervious study performed by us, which identified IAV in nasal swabs of piglets from three days of age [45]. In general, a high percentage of litters were positive for IAV in the farrowing unit, probably as a consequence of only two farrowing stables being available, meaning that new-born piglets were housed side-by-side with piglets ready for weaning at 4 weeks of age. In turn, this provided an optimal environment for IAV transmission, as new naïve individuals were readily available for infection. The relatively high percentage of sows (33.3%) that tested positive for IAV in nasal swabs in this study, suggested that the sows had an important role in the transmission dynamics. In addition, a significantly higher number of IAV positive sows also had an IAV positive litter, which suggested a transmission from sow-to-piglet or piglet-to-sow. These findings are in accordance with previous work [43,44,45,81], although more studies are needed to firmly determine the directionality of transmission between sows and piglets. The high number of sows found positive for IAV in this study and the fact that they were positive for approx. 1.5–2 weeks after being introduced into the farrowing unit, emphasize the importance of stimulating sow immunity, especially before entering an environment where IAV is circulating. If the sows are inadequately immunized when entering the farrowing unit, there is a high risk of IAV infection occurring a few days before birth, which potentially could lead to birth complications and a lower milking yield, which in turn will result in compromised animal welfare and production economy. First parity sows were overrepresented among the IAV positive sows, even though this correlation was not statistically significant. In the herd investigated here, the internally recruited gilts were kept in a quarantine stable for eight weeks and were thereby not exposed to the herd strain several weeks prior to re-introduction into the sow-herd. This could explain why the gilts seemingly were more prone to IAV, compared to the older sows continuously exposed to the herd strain. This further highlights that proper gilt immunization is important, through either natural exposure or vaccination. In summary, the results of the study suggest that the constant production of naïve piglets and the continuous introduction of gilts play a key role in the herd-level persistence of swIAV. 

The results of the study also highlighted the importance of herd management in the control of viral diseases. The same MCREBEL principles, which are widely used in controlling PRRSv infection [82], could also be a helpful tool in the prevention of IAV transmission within the herd. A clear sectioning and all in/all out management of weekly batches in the farrowing unit would most likely limit the IAV transmission significantly in the present herd. 

The overall mean coughing index (CI) was significantly higher in IAV positive litters/pens. This supports our previous findings [45] which found a significant correlation between the CI and the pen/litter testing positive for IAV in nasal swabs. In turn, CI could be a helpful tool in identifying IAV positive litters/pens for diagnostics. Furthermore, the correlation to clinical symptoms of respiratory disease underline that IAV has an impact on the health and welfare of infected pigs. 

Throughout the study, all eight genomic segments of the circulating H1avN2sw strain were highly similar (98.5–100%) to the initial consensus sequence, supporting that only a single IAV variant was circulating in the herd during the study. This is consistent with the information that the herd operated as a closed herd with no import of pigs, as the gilts were internally recruited. In turn, this provided the optimal scenario to study the viral drift within a single IAV strain, as the risk of reassortment events was limited. A correlation between sampling time and genetic diversity was found for all eight segments of the IAV strain, with the strongest temporal signal being present in the HA gene. The phylogenetic tree based on HA amino acid sequences had the distinct comb-like topology also known from human HA sequences, with a main trunk that represents the pathway of advantageous (mostly immune escape) mutations, which have been selected over time. Conversely, the shorter side branches represents the isolates that died out, because they were not able to avoid the host immunity [16,83,84,85]. Likelihood-based analysis using CODEML, confirmed the presence of positive selection in the HA gene. Further inspection of the HA amino acid alignment showed how amino acid mutations at specific positions arose at a given time point and remained until the end of the study, consistent with these amino acid changes being advantageous. All the mutations manually identified corresponded to positions in which the dN/dS ratio was estimated to be higher than the average (using Bayesian analysis). Moreover, the CODEML analysis defined the same mutations to be ancestral amino acid changes (presented in Figure 2), thereby supporting the presence of repeated bottlenecks. However, the dN/dS ratios remained below 1, probably due to a low number of sequences obtained at each sampling. Several of these positions were located in previously known antigenic sites (Sa, Sb, Ca1, Ca2, and Cb) or in other known B or T-cell epitopes, suggesting that positive selection occurred in immunogenic important sites. The single positively selected site (dN/dS ratio > 1) identified by the CODEML and Bayesian analysis, was located in the HA2 subunit of the HA gene, which encodes the stalk region. Interestingly, this position was included in a B-cell epitope proposed by a previous study investigating the A(H1N1)pdm09 subtype. In this study the reaction of the peptide encoding the epitope against a panel of swine sera raised against a panel of H1 and H3 subtypes were tested and showed a reaction, indicating this epitope to be highly immune reactive in pigs [58]. Moreover, the position was also included in a T-cell epitope defined in the human seasonal H1N1 subtype [59]. The results of the two studies indicate that the positive selection identified in position 536, could have an impact on the immunity, and thereby be important for immune escape variants of swIAV. However, this specific amino acid change did not show the same evolutionary bottleneck pattern as the other sites with elevated dN/dS ratios, as the change appeared at several independent sampling. 

The nucleotide substitution rate calculated for the HA gene in this study exceeded the nucleotide substitution rate of 1.9 to 4.37 × 10^−3^ substitutions per site per year for swine H1Nx subtypes reported in previous studies [11,30,31,32,33]. The nucleotide substitution rate was also markedly higher than that reported for human H1N1 [11,86,87]. However, it is a well-known phenomenon, that when sequentially viral samples are collected over a short period of time, the clock rate is often estimated to be much higher compared to values estimated from samples collected over a longer period of time [88,89,90]. Therefore, it is difficult to compare the within-herd substitution rate calculated in this study, with the overall substitution rates calculated previously based on samples collected over several years and originating from many different herds or people. However, the nucleotide substitution rate of the NP gene corresponds to the results of an earlier study [91]. The average dN/dS (global ω ratio) of the HA sequences of this study was similar to that found in two studies investigating human H1N1 subtypes (ω ratio: 0.18–0.21) [32,92], but lower than that documented in two other studies (ω ratio: 0.24–0.38) [11,12]. The results of our study need to be confirmed by additional within-herd studies of the viral evolution, however, the data obtained in the present study suggested that the within herd evolution of swine IAV are comparable to that of human seasonal IAV. The rate of nucleotide substitution was high, and there was clear evidence for positive selection on the HA gene, especially in epitopes important for the adaptive and cellular immune response. Moreover, the topology of the phylogenetic tree indicated that immune escape variants were selected over time. Consequently, the results further confirmed that the antigenic drift of IAV in swine is comparable to seasonal human IAV. The extensive antigenic drift with generation of escape variants could possibly lead to acute clinical swIAV outbreaks even in herds without an external IAV introduction. In addition, the intensive viral drift could also have a negative impact on vaccine efficacy, however, this need to be studied in more detail. Nevertheless, it is recommendable for all herds to get their IAV strain/s sequenced. Then, the herd veterinarian will have an additional tool in explaining disease developments and it will contribute to our general understanding of the viral evolution of IAV in swine. From a human health point of view, it is also important to understand the viral evolution of IAV in swine herds, as they represents a reservoir for generation of future human pandemics, as demonstrated by the 2009 human pandemic [93,94]. The risk of generating new human pandemics will probably increase as an increasing number of herds become persistently infected, keeping a constant evolving reservoir of IAV circulating in swine. 

## 5. Conclusions

In conclusion, the present study confirmed other recent studies [40,44,46,81,95] that found that swIAV infections should be regarded as enzootic infections with long term within-herd persistence. Our results also revealed that this change in epidemiology potentially affected the viral evolution, measured as increased diversity and selection of escape mutants. Finally, persistent circulation of swIAV in swine herds increase the likelihood for generation of re-assortments between human and swine IAV strains.

## Figures and Tables

**Figure 1 viruses-12-00248-f001:**
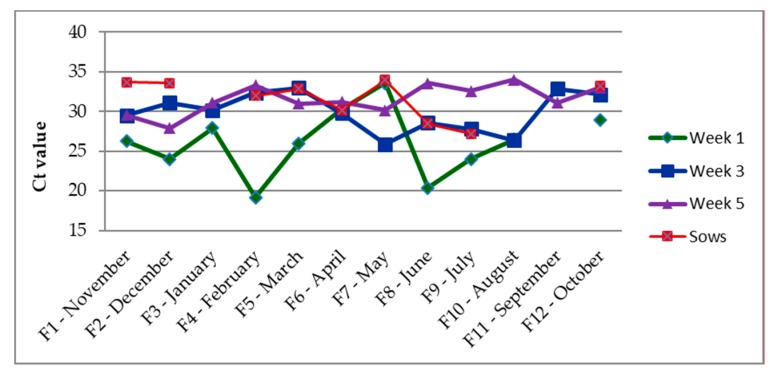
The average Ct value of the influenza A viruses (IAV) positive litters/pens and sows at each sampling time.

**Figure 2 viruses-12-00248-f002:**
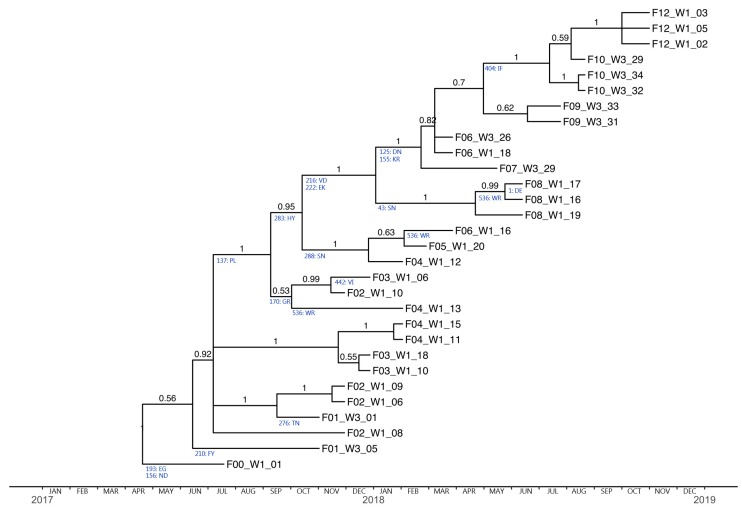
Bayesian strict molecular clock tree of the hemagglutinin (HA) sequences. The x-axis represents time in months from 2017–2019. Branch labels represent posterior clade probabilities. Ancestral amino acid changes are indicated below the branches on which they most probably occurred, with DTLC numbering. F00_W1_01 sampled approx. four months before the actual study was used as an outgroup.

**Figure 3 viruses-12-00248-f003:**
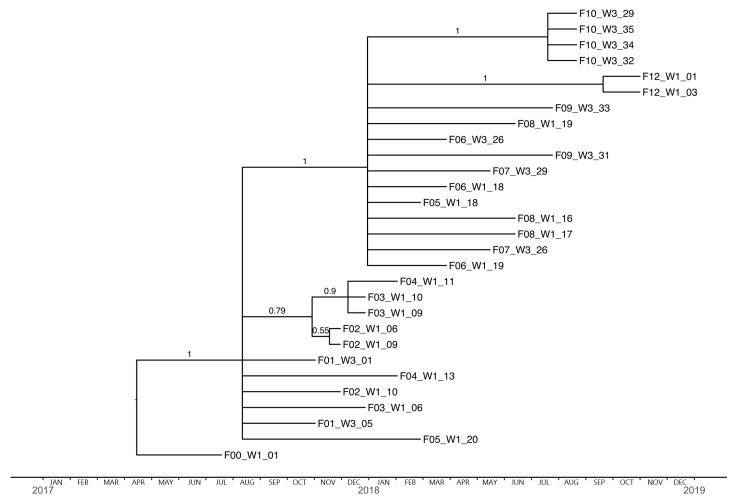
Bayesian strict molecular clock tree of the NA sequences. The x-axis represents time in months from 2017–2019. Branch labels represent posterior clade probabilities. F00_W1_01 sampled approx. four months before the actual study was used as an outgroup.

**Table 1 viruses-12-00248-t001:** Prevalence of swine influenza A viruses (swIAV) positive litters/pens and sows at each sampling time.

	Week 1	Week 3	Week 5	Sows
Nov	75% (3/4)	100% (4/4)	75% (3/4)	50% (2/4)
Dec	75% (3/4)	75% (3/4)	25% (1/4)	50% (2/4)
Jan	100% (4/4)	75% (3/4)	100% (4/4)	0% (0/4)
Feb	75% (3/4)	50% (2/4)	100% (4/4)	75% (3/4)
Mar	75% (3/4)	75% (3/4)	50% (2/4)	100% (4/4)
Apr	50% (2/4)	75% (3/4)	75% (3/4)	25% (1/4)
May	100% (4/4)	50% (2/4)	75% (3/4)	25% (1/4)
Jun	25% (1/4)	25% (1/4)	25% (1/4)	25% (1/4)
Jul	25% (1/4)	100% (4/4)	75% (3/4)	25% (1/4)
Aug	50% (2/4)	100% (4/4)	50% (2/4)	0% (0/4)
Sep	0% (0/4)	50% (2/4)	50% (2/4)	0% (0/4)
Oct	75% (3/4)	50% (2/4)	25% (1/4)	25% (1/4)
Total	60.5% (29/48)	70.8% (34/48)	60.5% (29/48)	33.3% (16/48)

**Table 2 viruses-12-00248-t002:** Average coughing index (CI) in IAV positive and negative litters/pens.

	Week 1	Week 3	Week 5	Total
IAV Positive	0.12 (SD = 0.14)	0.36 (SD = 0.27)	0.09 (SD = 0.08)	0.32 (SD = 0.23)
IAV Negative	0.05 (SD = 0.08)	0.30 (SD = 027)	0.06 (SD = 0.04)	0.12 (SD = 0.17)
*p*-value	0.07	0.48	0.12	0.03

**Table 3 viruses-12-00248-t003:** The best fitting substitution model, temporal correlation coefficient and nucleotide substitution rate of each gene segment.

	M1a		M2a		Significant Difference	Correlation Coefficient	Substitution Rate
	AIC	Akaike Weight	AIC	Akaike Weight	*p*-Value		
HA	5494.30	0.1143	5490.20	0.8857	<0.05	0.95	7.6 × 10^−3^
NA	4203.24	0.9478	4209.06	0.0521	>0.05	0.81	6.9 × 10^−3^
M	2866.58	0.8810	2870.58	0.1189	>0.05	0.82	2.5 × 10^−3^
NS	2531.02	0.8810	2535.02	0.1189	>0.05	0.68	5.7 × 10^−3^
NP	4310.14	0.8810	4314.14	0.1189	>0.05	0.77	1.1 × 10^−3^
PB1	6595.06	0.8810	6599.06	0.1189	>0.05	0.86	2.47 × 10^−3^
PB2	6459.84	0.8810	6463.84	0.1189	>0.05	0.94	2.94 × 10^−3^
PA	6170.36	0.8810	6174.36	0.1189	>0.05	0.60	2.13 × 10^−3^

M1a = model 1, which describes neutral and negative selection. M2a = model 2, which describes neutral, negative and positive selection. AIC = Akaike information criterion. The correlation coefficient of the TempEst analysis. The substitutions rate gives the results of the BEAST analysis expressed as nucleotide substitution rate per site per year.

**Table 4 viruses-12-00248-t004:** Evidence of positive selected amino acid changes and their location in antigenic sites.

Position (from DTLC)	Mutation	dN/dS Ratio	PR+	Sequences Showing the Mutation	Antigenic Site/Epitope/Glycosylation Site
1	D→E	0.7982	0.2721	1/3 F8	T-cell
43	S→N	0.8255	0.2837	3/3 F8	B-cell
125	D→N	0.7816	0.2651	2/3 F6, 1/1 F7 and all F9-F12	Sa
137	P→L	0.8791	0.3065	1/4 F2, 1/3 F3, 2/4 F4 and all F5–F12	Ca1
155	K→R	0.7672	0.2588	2/3 F6, 1/1 F7 and all F9–F12	Sa
156	N→D	0.7991	0.2725	All F1-F12	Ca1
170	G→R	0.8784	0.3062	1/4 F2 and 1/3 F3	Ca1
193	E→G	0.8746	0.3046	All F1–F12	Sb
210	F→Y	0.8215	0.2820	1/2 F1	B-cell epitope
216	V→D	0.8407	0.2903	2/3 F6 and all F7–F12	B-cell epitope
222	E→K	0.7717	0.2607	2/3 F6 and all F7–F12	Ca2
276	T→N	0.8282	0.2849	1/2 F1	Glyco
283	H→Y	0.8402	0.2898	1/4 F4 and all F5–F12	-
288	S→N	0.8255	0.2838	1/4 F4, 1/1 F5 and 1/3 F6	-
404	I→F	0.8102	0.2773	All F10–F12	T-cell epitope
442	V→I	0.8036	0.2745	1/3 F3	T-cell epitope
536	W→R	**3.156** 1.5882	**0.879** 0.6380	1/4 F4, 1/3 F6 and 2/3 F8	B-cell and T-cell epitope

Column 1 indicates the position with an elevated dN/dS ratio identified by the Bayesian analysis (normal text) and in both the Bayesian analysis and CODEML (bold text). Column 2 indicates the amino acid change occurring over time. Column 3 and 4 gives the dN/dS ratio and the probability (PR+) of the position being positive selected with the results of the Bayesian analysis in normal text and the CODEML results in bold text. Column 5 presents the sequences wherein the given mutation was identified; herein F1–F12 indicates the sequences where the given mutation was identified. Column 6 specifies if the mutation was located in an antigenic site (Sa, Sb, Ca1, Ca2, and Cb) [5,6,8,55,56], glycosylation site (Glyco) [63] or a B-cell [57,58] or T-cell epitope [59,60,61,62].

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
