# Peer review of "Substantial Antigenic Drift in the Hemagglutinin Protein of Swine Influenza A Viruses"

_viruses, 2020, doi:10.3390/v12020248_

Round 1

Reviewer 1 Report

The manuscript of Ryt-Hansen describes experiments conducted to investigate the antigenic drift of swine influenza A viruses (swIAV). The authors selected a pig farm with enzootic swIAV infection of pigs. Five nasal swabs each of 4 litters of the following age groups have been collected monthly for 1 year: 1-week-old piglets, 3-week-old piglets and 5-week-old nursery pigs and of the 4 sows of the 1-week-old piglets. The prevalence of swIAV was determined as well as the coughing index of litters/pens. Full genome sequences of ten cultured isolates plus HA and NA sequences of unculturable samples were obtained and used for phylogenetic analysis and determination of substitution rates and positive selection. As a result, the authors observed a correlation of swIAV infection and coughing index, observed genetic drift and obtained signals suggesting positive selection of the HA gene.

This is largely a thorough and interesting study, but parts of the discussion are highly speculative which either need experimental support or should be toned down. Further, this manuscript needs few corrections.

line 22: It should read enzootically infected swine herd. line 27 and elsewhere in the text: It should read substitution rather than mutation. A mutation is usually associated with an altered phenotype. line 47: It should read "Several antigenic sites have been identified...". lines 57-60, 383-385: It appears that a recent study by Zell et al. (Arch Virol 165:55-67, 2020) came to similar substitution rates. This should be discussed. line 88: It should read Sarcoptes scabiei. line 211: It should read "... and the Ct values of the samples did not allow...". lines 244-245, 254: There seems to be a mismatch regarding the numbers of HA and NA sequences that were generated and submitted to GenBank and shown in the phylogenetic trees (Figs. 2 and 3). Please explain. lines 288ff.: The authors describe a W-to-R substitution at aa position 553 in 4 pigs of the F4, F6, F8 samples. Please specify the exact sample numbers. When comparing the HA tree, this substitution must have emerged several times independently which questions the bottleneck hypothesis. I was wondering why positive selection was not detected in the predominant antigenic sites. How do the authors know that B-cell or T-cell epitopes which had been identified in pandemic viruses and the ceased human seasonal viruses are also present in swIAV? Is there sequence similarity and if so is this site functional? Table 4: This table presents "positive selected amino acid changes". Per definition, positive selection (diversifying selection) requires dN/dS values greater 1. All but 1 sites have dN/dS ratios smaller 1. To me it appears that these are sites with less stringent negative selection. Could undersampling explain this. Would more monthly samples or an extended sampling period improve dN/dS ratios? The conclusions should be weakened in the text. line 365: It should read "side branches". lines 362-366: If the "main trunk" of the phylogenetic tree represents advantagous aa exchanges, the authors are advised to indicated these substitutions at the respective nodes of the tree. This would also support their hypothesis of bottlenecks. Figures 2 and 3: The scale of the x-axis should present the months (Nov 2017, Dec 2017, Jan 2018, etc.) rather than "time in years". It is uneasy for the reader to convert the present scale to the relevant dates.

Author Response

Response to reviewers:

Thank you for your review and very useful comments that indeed will improve the paper. We have tried to address and incorporate all of your comments as described in details below:

Reviewer 1:

The manuscript of Ryt-Hansen describes experiments conducted to investigate the antigenic drift of swine influenza A viruses (swIAV). The authors selected a pig farm with enzootic swIAV infection of pigs. Five nasal swabs each of 4 litters of the following age groups have been collected monthly for 1 year: 1-week-old piglets, 3-week-old piglets and 5-week-old nursery pigs and of the 4 sows of the 1-week-old piglets. The prevalence of swIAV was determined as well as the coughing index of litters/pens. Full genome sequences of ten cultured isolates plus HA and NA sequences of unculturable samples were obtained and used for phylogenetic analysis and determination of substitution rates and positive selection. As a result, the authors observed a correlation of swIAV infection and coughing index, observed genetic drift and obtained signals suggesting positive selection of the HA gene.

This is largely a thorough and interesting study, but parts of the discussion are highly speculative which either need experimental support or should be toned down. Further, this manuscript needs few corrections.

line 22: It should read enzootically infected swine herd.

  • This has now been changed L.22

line 27 and elsewhere in the text: It should read substitution rather than mutation. A mutation is usually associated with an altered phenotype.

  • We use the word “substitution” throughout the paper to describe nucleotide changes and “mutations” to describe amino acid changes, and therefore it would be confusing for the reader to change the word mutation to substitution in this case.

line 47: It should read "Several antigenic sites have been identified...".

  • This has now been changed L. 47

lines 57-60, 383-385: It appears that a recent study by Zell et al. (Arch Virol 165:55-67, 2020) came to similar substitution rates. This should be discussed.

  • Thank you for the observation, we have now added and discussed the paper L.62-65.

line 88: It should read Sarcoptes scabiei.

  • This has now been corrected L.91-92.

line 211: It should read "... and the Ct values of the samples did not allow...".

  • This has now been corrected L.221.

lines 244-245, 254: There seems to be a mismatch regarding the numbers of HA and NA sequences that were generated and submitted to GenBank and shown in the phylogenetic trees (Figs. 2 and 3). Please explain.

  • This is now explained L.266

lines 288ff.: The authors describe a W-to-R substitution at aa position 553 in 4 pigs of the F4, F6, F8 samples. Please specify the exact sample numbers. When comparing the HA tree, this substitution must have emerged several times independently which questions the bottleneck hypothesis.

  • This has now been specified L. 311.,and a paragraph in the discussion now mentioned that exactly this mutation does not behave like the other mutations as it appears in several unrelated samplings L404-405.

I was wondering why positive selection was not detected in the predominant antigenic sites.

  • They do, as indicated in table 4, almost all amino acid changes are occurring in antigenic sites. However, the dN/dS rate is only elevated and not above 1, which might be due to a low number of samples at each sampling time. This is now discussed in L.393-394.

How do the authors know that B-cell or T-cell epitopes which had been identified in pandemic viruses and the ceased human seasonal viruses are also present in swIAV? Is there sequence similarity and if so is this site functional?

  • We cannot know that for sure. However, no other data is available for swIAV yet, and other publications also refers to what has been documented in huIAV. Moreover, the sequence similarities between swine pandemic H1N1 and human pandemic H1N1 is quite high.

Table 4: This table presents "positive selected amino acid changes". Per definition, positive selection (diversifying selection) requires dN/dS values greater 1. All but 1 sites have dN/dS ratios smaller 1. To me it appears that these are sites with less stringent negative selection. Could undersampling explain this. Would more monthly samples or an extended sampling period improve dN/dS ratios? The conclusions should be weakened in the text.

  • The title of table 4 has now been changed to “Evidence of positive selected amino acid changes and their location in antigenic sites”, and the caption also now reads “Column 1 indicates the position with an elevated dN/dS ratio”. Moreover, you are right; the low sample number is probably the reason for dN/dS rates below 1. However, they are still significantly elevated compared to other sites and corresponds to the ancestral amino acid changes identified by CODEML. This is now discussed in L. 391-394.

line 365: It should read "side branches".

  • This has now been changed L. 385.

lines 362-366: If the "main trunk" of the phylogenetic tree represents advantagous aa exchanges, the authors are advised to indicated these substitutions at the respective nodes of the tree. This would also support their hypothesis of bottlenecks.

  • The amino acid changes occurring at the different nodes in the tree has now been added to Figure 2 and corresponds to the mutations listed in Table 4, thereby supporting the bottleneck theory. The method for this is described in L194-198.

Figures 2 and 3: The scale of the x-axis should present the months (Nov 2017, Dec 2017, Jan 2018, etc.) rather than "time in years". It is uneasy for the reader to convert the present scale to the relevant dates.

  • The time scale has now been changed in Figure 2 and 3 and the method is described in L. 205-207.

Reviewer 2 Report

The authors monitored influenza virus circulation dynamics in a swine herd throughout a year. They found sustained virus circulation within the herd and showed by sequencing that the strain acquired mutations over time at a similar substitution rate as observed in humans. It is an interesting study, but it would be helpful if some points could be discussed in more detail.

The herd contains "only" approximately 2500 animals and the authors state that no new animals were introduced throughout the study period (which presumably also means no animals were born?). This seems like a small cohort to support continuous influenza virus circulation. I admittedly have more knowledge about human influenza virus circulation, but in general it seems like 25-50% of sows are constantly infected. I did not see any reference to specific testing of symptomatic animals, so this seems like a very high rate at which the virus should quickly "burn" through the population. Is this not common within swine? Similarly, outside of tropical countries, influenza virus circulation is substantially lower during summer months, but seemed still fairly high in this study. Do the same restrictions and dynamics not apply to swine? It would be interesting if the authors could comment on what this may mean for what is generally thought to be a physical limitation of virus stability.

Author Response

Response to reviewers:

Thank you for your review and very useful comments that indeed will improve the paper. We have tried to address and incorporate all of your comments as described in details below:

Reviewer 2:

The authors monitored influenza virus circulation dynamics in a swine herd throughout a year. They found sustained virus circulation within the herd and showed by sequencing that the strain acquired mutations over time at a similar substitution rate as observed in humans. It is an interesting study, but it would be helpful if some points could be discussed in more detail.

The herd contains "only" approximately 2500 animals and the authors state that no new animals were introduced throughout the study period (which presumably also means no animals were born?). This seems like a small cohort to support continuous influenza virus circulation. I admittedly have more knowledge about human influenza virus circulation, but in general it seems like 25-50% of sows are constantly infected. I did not see any reference to specific testing of symptomatic animals, so this seems like a very high rate at which the virus should quickly "burn" through the population. Is this not common within swine?

  • The herd have weekly farrowings – which means new naïve individuals being born weekly. So the number of animals indicate how many animals are present on a specific day/how many pigs can be in the stables. However, the individual animals are replaced very quickly. I have now specified L.81-89 so that this should be explained. And added a section to the discussion that emphasize that it is indeed the high replacement rate of gilts and constant production of piglets that keep the circulation of IAV going L.362-364.

Similarly, outside of tropical countries, influenza virus circulation is substantially lower during summer months, but seemed still fairly high in this study. Do the same restrictions and dynamics not apply to swine? It would be interesting if the authors could comment on what this may mean for what is generally thought to be a physical limitation of virus stability.

  • Swine IAV does not have the same seasonal circulation pattern as human influenza, and therefore this study confirms the results of previous studies and a section in the discussion has been added to clarify this L. 336-337.